# ONLINE WEIGHT APPROXIMATION FOR CONTINUAL LEARNING

## ABSTRACT

Continual Learning primarily focuses on studying learning scenarios that challenge a learner's capacity to adapt to new problems, while reducing the loss of previously acquired knowledge. This work addresses challenges arising when training a deep neural network across numerous tasks. We propose an Online Weight Approximation scheme to model the dynamics of the weights of such a model across different tasks. We show that this represents a viable approach for tackling the problem of catastrophic forgetting both in domain-incremental and class-incremental learning problems, provided that the task identities can be estimated. Empirical experiments under several configurations demonstrate the effectiveness and superiority of this approach also when compared with a powerful replay strategy.

## 1 INTRODUCTION

Continuously learning from an evolving source of data rather than from a fixed dataset has been one of the most compelling and unsolved problem in the deep learning landscape. Indeed, standard learning methods for Deep Neural Networks (DNNs) struggle when they keep learning on data drawn from a new distribution while preserving knowledge about previously encountered examples. This phenomenon is known as *catastrophic forgetting* (McClelland et al. (1995); McCloskey & Cohen (1989)) and it entails that DNNs are unable to correctly process examples coming from past distributions. As one can imagine, this is a major limitation of current artificial intelligence method, since the capacity to continuously learn might be a necessity in different scenarios. As an example, we may need to learn to recognize new samples belonging to an already known class (domain-incremental learning); in other cases, we may want our model to extend the set of recognized classes without losing the capability to predict the ones previously learnt (class-incremental learning) Mai et al. (2022).

Several approaches have been developed to tackle this problem either by inserting regularisation mechanisms which prevent the network from abruptly shifting parameters when facing a new learning distribution Kirkpatrick et al. (2017); Ahn et al. (2019), or architectural constraints which aim to specialize different parameters in learning different distributions Rusu et al. (2016); Xu & Zhu (2018), or even memory-based approaches which retain and keep learning over a set of samples from the past distribution Rebuffi et al. (2017); Rolnick et al. (2019). While all these approaches are useful to mitigate it, the *catastrophic forgetting* problem is far from being solved, and it remains an important challenge in the deep learning research field.

Inspired by recent developments in online approximation of functions Voelker et al. (2019); Gu et al. (2020) here we propose to capture the dynamic of the parameters of a DNN model by an approximation function that is efficiently updated online. By means of this function, we can retrieve a representation of the weights at any given moment of time. As a result, we retain the capability to predict over samples coming from the past. In the experiments, we show that when employing a sufficiently powerful approximation function, the loss of performance with respect to a model trained on the past distribution only is negligible. Moreover, when comparing the proposed method with respect to a replay strategy equipped with a similar budget, we increase the average accuracy by up to $+34\%$ points in the tested configurations.

As previously mentioned, in this work we concentrate on scenarios where at test time the identifier of the distribution from which the sample was collected is predetermined or known in advance.

This assumption forms the basis of several existing works, such as Lopez-Paz & Ranzato (2017); Li & Hoiem (2016) and it holds true in scenarios where contextual information about the sample is either available or can be predicted, either by the model itself or by an external source. Consider, for instance, the case in which a sensor gradually corrupts acquired data over time. The age of the sensor serves as valuable information, and it is accessible both during training and testing phases. Similarly, in cases where the recognition of a set of objects depends on the specific scene they are placed in, this contextual information can either be readily available or, in some instances, predicted by the model.

To summarize, the main contributions of our approach are: (i) We propose for the first time an Online Weight Approximation Method (OWA) for neural model to tackle continual learning problems. (ii) We demonstrate that our proposed method enables achieving an average accuracy that matches the performance attained when training on a single task, in both continuous domain-incremental and class-incremental problems. (iii) We show that the proposed method allows to reach classification performance that are higher than an effective continual learning strategy such as Replay Buffer (up to $+34\%$ points in average accuracy).

## 2 BACKGROUND

**Online Approximation of Functions**  Given a function of one variable defined on the half line $u \colon [0, +\infty) \to \mathbb{R}$, the problem of online approximation of such function is twofold: *i.* for time instant $t \in [0, +\infty)$ find an approximation of $u$ until $t$, i.e. $u^t := u|_{I_t}$ with $I_t := (0, t)$ and *ii.* have a method to update online such approximation. In order to define a notion of approximation, we need to have some notion of closeness, and hence we need to assume that the function that we want to approximate lives in some normed space. Here we only discuss the case of square integrable functions. The measure with respect to which we define the notion of integrability has a rather important role for computing an online approximation (for a more detailed account on this point see Gu et al. (2020)). In this work, we simply assume to work with a normalized Lebesgue measure on $I_t$ (which is the standard choice in $\mathbb{R}^n$). The other basic ingredient is the class of basis functions with which we want to perform the approximation. In this setting, Gu et al. (2020) have shown that a good choice is represented by the translated and rescaled Legendre polynomials $v_n^t$ for $n = 1, 2, \ldots$ defined in $[0, t]$ by

$$v_n^t(x) = \sqrt{2} e_n \left( \frac{2x}{t} - 1 \right) \quad \forall x \in [0, t], \quad n = 0, 1, \ldots, \tag{1}$$

where $e_n$ are the normalized Legendre polynomials (see Ciarlet (2013)). Then, the wanted approximation $v^t$ of the function $u^t$ can be expressed (as explained in Gu et al. (2020)) by

$$v^t = \sum_{n=0}^{N-1} c_n(t) v_n^t \quad \text{where} \quad c_n(t) := (u^t, v_n^t)_t, \tag{2}$$

where $(u^t, v_n^t)_t := \int_{I_t} u^t v_n^t \, dx/t$ is the standard scalar product in $L^2((0, t); \mathbb{R})$ rescaled by a factor $1/t$. This is a very well known and general results that is at the basis of Fourier theory and gives an answer to the approximation problem *i.* stated above. Recently, however, inspired by a previous work on Legendre polynomials Voelker et al. (2019), Gu et al. (2020) have shown that this approach to online approximation offer a particularly efficient (linear) way of updating such approximation online. The basic idea is that once we extend $u^t$ to the whole half-line $\mathbb{R}_+$ by identifying $u^t$ with $u 1_{I_t}$ ($1_A$ being the characteristic function of the set $A$) we realize that the coefficients $c_n$ defined in equation 2 are differentiable in the classical sense. In particular, the main result that we are going to use can be summed up in the following theorem:

**Theorem 2.1** (Gu et al. (2020)). *Let $u \in \mathcal{C}^0([0, +\infty), \mathbb{R})$ and let $v_t^n$ be as in equation 1. Let*

$$c_n(t) = (u^t, v_n^t)_t = \frac{1}{t} \int_{(0, +\infty)} u(x) v_n^t(x) \, 1_{I_t}(x) dx,$$

*then $c_n \in \mathcal{C}^1([0, +\infty), \mathbb{R})$ for all $n = 0, \ldots, N$ and, in particular*[1]

$$\dot{c}(t) = -\frac{1}{t} A c(t) + \frac{1}{t} B u(t), \tag{3}$$

---

[1] Here we use the notation $\dot{c}$ to represent the derivative of the function $t \mapsto c(t)$.

*where $c(t) = (c_0(t), \ldots, c_{N-1}(t))$ and*

$$A_{ij} = \begin{cases} \sqrt{(2i+1)(2j+1)} & \textit{if } i > j \\ i+1 & \textit{if } i = j \\ 0 & \textit{otherwise} \end{cases}, \quad B_i = \sqrt{2i+1} \quad \textit{for } i, j = 0, \ldots, N-1. \quad (4)$$

Equation 3 allow us to update the approximation online as follows: suppose that we have computed for some $t_0 \in (0, +\infty)$ the coefficients $c^0$ according to equation 2, then the coefficients of the approximation at some later time $t_1 > t_0$ can be computed by solving the Cauchy problem given by equation 3 with initial condition $c(t_0) = c^0$ up to time $t_1$ instead of using again equation 2.

**Continual Learning** Continual Learning is mostly interested in the problem of learning neural models from a stream of data. The sequentiality of the problem is reminiscent and partly inspired on the way in which humans learns. The main challenge in doing this is represented by the problem of forgetting, or as it is known in the literature, *catastrophic forgetting*.

Throughout this paper we will assume the more definite scenario of *task incremental learning* (see e.g. De Lange et al. (2022)) in which we will assume that the sample from which we want to learn are collected in homogeneous groups that we process sequentially; in other words we will assume that we have a stream of tasks. For the purposes of this work we will not give a precise definition of task, and we will simply adopt the working definition given in De Lange et al. (2022) of *an isolated training phase with a new batch of data, belonging to a new group of classes, a new domain, or a different output space*. The method that we are going to propose works in a scenario in which, at inference time, we are able to retrieve the information on the task, so we will assume that either the ID of the task is available or that we are able to infer it.

## 3 METHOD

Our proposal is to apply the online function approximation method described in Section 2 to the parameters (weights) of a neural network in order to build a compact representation of the dynamics of the weights across different tasks. Let us consider a a DNN $f(\cdot, w): \mathbb{R}^d \to \mathbb{R}^m$ with weights $w \in \mathbb{R}^p$. Assume furthermore that we are in a continual learning scenario with $T \gg 1$ tasks and that for each task $t$ we are able to find a set of weights $w^t$ that solves the learning problem in that specific task. A learning procedure in this setting will have as as outcome the sequence $w^1, w^2, \ldots, w^T$ of the learned weights after each task.

Assuming that at inference time we are able to retrieve the for a specific example the corresponding task ID $\bar{t}$, our aim would be to restore the set of weights $w^{\bar{t}}$ instead of the final weights $w^T$. Doing this would require to keep in memory all the weights found for all the $T$ tasks and therefore would require a memory budget that is proportional to $pT$ ($p$ being the number of parameters of the model). Instead of storing all the weights for all the tasks, the approximation method described in Section 2 allows us to approximate the weight trajectory with a constant amount of memory (depending only on the order of the approximation $N$) that does not increase with the number of tasks. First, however, we need to discuss the following issues: *i.* how do we deal with an approximation of a sequence of vectors (the weights) instead of a scalar-valued function and *ii.* how can we address in this discrete setting the hypothesis of continuity of the approximated function as required in Theorem 2.1.

We chose to solve issue *i.* simply by applying the online approximation procedure to each components of the weights separately, and we simply apply Theorem 2.1 that holds for scalar functions to each vector component. Doing this we turn the problem into $p$ independent online approximation problems one for each weight of the network as it is depicted in Fig. 1. Such approach then boils down to the determination of the trajectories of the vectors of coefficients $t \mapsto C_i(t) \in \mathbb{R}^N$ over time, one for each weight $w_i$ of the network.

To address *ii.* instead, we need to give a discrete interpretation of equation 3, that is also what our implementation will use. Let us focus from now on the $i$-th weight $w_i$. Clearly, given a sequence of values of a given weight $w_i^1, w_i^2, \ldots$, they can always be interpreted as the *sampling* of some continuous function done with some temporal resolution $\tau > 0$. In other words, we can always find at least a continuous function $\omega_i: \mathbb{R}_+ \to \mathbb{R}$ such that $\omega_i(k\tau) = w_i^k$ and therefore we can apply

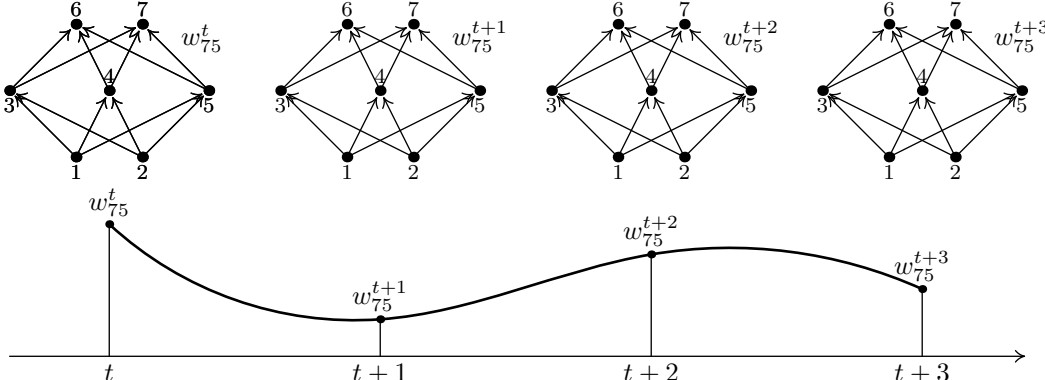

Figure 1: Conceptual illustration of how the OWA method works. For each weight of a network (in this case we are focusing our attention on the weight $w_{75}$), we use an online approximation scheme to model the dynamics of the weights across different tasks and restore previous configuration at test time.

the approximation method to the family of functions $\omega_i$ for $i = 1, \ldots p$. This means that the set of coefficient $C_i(t)$, for each $i = 1, \ldots, p$ satisfies:

$$\dot{C}_i(t) = -\frac{1}{t}AC_i(t) + \frac{1}{t}B\omega_i(t), \quad i = 1, \ldots, p. \tag{5}$$

Notice that each vector of coefficients $C_i$ is computed using the same constant matrices defined in equation 4. Still, when we are working in a discrete setting, we need to solve the main equation 5 only knowing a sampling $w_i^1, w_i^2, \ldots$ of the term $\omega_i$. As we will see, this make the issue of continuity of the weight trajectory that we have just discussed closely intertwined with the precision of approximation technique that we are going to use to solve the ODE system 5 that we are going to discuss now.

**Temporal Discretization** We will assume to numerically solve equation 5 using a straightforward Euler's method. This means that given a temporal quantization step $\tau$, we can define a partition of the half line $(0, +\infty)$, $\mathcal{P} := \{0 = t_\tau^0 < t_\tau^1 < \cdots < t_\tau^n < \ldots\}$ with $t_\tau^n = t_\tau^{n-1} + \tau$. The sequences of vectors $(\tilde{C}_i^k)_{k=0}^{+\infty}$ that we expect to be an approximation of $C_i(t_\tau^k)$, one for each weight $w_i$, is defined by the following recursive relation (explicit Euler method):

$$\frac{\tilde{C}_i^{k+1} - \tilde{C}_i^k}{\tau} = \frac{1}{k\tau}\left(-A\tilde{C}_i^k + Bw_i^k\right) \quad i = 1, \ldots, p \quad \text{and} \quad k > 0, \tag{6}$$

that can be rewritten more explicitly as

$$\tilde{C}_i^{k+1} = \left(\text{Id} - k^{-1}A\right)\tilde{C}_i^k + k^{-1}Bw_i^k. \quad i = 1, \ldots, p \quad \text{and} \quad k > 0, \tag{7}$$

Notice that equation 7 is independent of $\tau$; this is a consequence of the equivariance property of ODE system for the coefficients in continuous time that can be summed up in the following statement:

**Proposition 3.1** (proposition 3 of Gu et al. (2020)). *Let $\alpha > 0$ and define $\Phi \colon [0, +\infty) \to [0, +\infty)$ that maps $t \mapsto s = \alpha t =: \Phi(t)$ and let $c$ be a solution of equation 3, then if we define $\hat{c}(s) = c(\Phi^{-1}(s)) = c(t)$ we have that $\hat{c}$ solves $\dot{\hat{c}}(s) = -s^{-1}A\hat{c}(s) + s^{-1}Bu(s/\alpha)$.*

This result basically means that if we rescale the time of the function $u$ by a factor $1/\alpha$, then the dynamics of the coefficients will be rescaled by the same factor. This property is instrumental to make us understand the goodness of the approximation of the Euler method in equation 7 in relation to the "regularity" properties of the sequence of weights $w_i^1, w_i^2, \ldots$. Indeed, since we want to interpret $w_i^1, w_i^2, \ldots$ as a sampling of a continuous function, as we reduce the value of $\tau$ the sequence of coefficients $\tilde{C}_i^1, \tilde{C}_i^2, \ldots$ can be regarded as good approximation of the values $C_i(k\tau)$ only if the difference between $|w_i^{k+1} - w_i^k|$ is sufficiently small, namely of the same order of the

---

**Algorithm 1** Continual Learning with OWA

---

**Require:** A neural model $f$, a sequence of datasets $D_1, \ldots, D_T$ where dataset $D_j$ is of size $M_j$, a number of epochs $E_j$ for each task and an update policy $\pi_j^\alpha$ for $j = 1, \ldots T$ and $\alpha = 1, \ldots E_j$ and the order of desired approximation $N$.

**Initialize:** Randomly initialize the weights of the network $f$ and set $\tilde{C}_i = 0 \in \mathbb{R}^N$ for all $i = 1, \ldots, p$.

  **for** $j = 1$ upto $T$: **do**

    **for** $\alpha = 1$ upto $E_j$ **do**

      Update the parameters of the network $w$ using SGD

      **if** $\pi_j^\alpha = 1$ **then**

        Update $\tilde{C}_i$ for all $i = 1, \ldots, p$ using equation 7

      **end if**

    **end for**

  **end for**

---

$\tau$. [2] This means that even if we can indeed use equation 7 for a generic sequence of points $(w_i^k)_{k>0}$, the resulting sequence $(\tilde{C}_i^k)_{k>0}$ will be a good approximation of equation 5 only when $(w_i^k)_{k>0}$

1. it comes from a truly continuous underlying process and

2. it has been sampled at an appropriate resolution.

These two points are important guideline for using the online approximation method in a continual learning scenario since, as we will discuss in the experimental section, the rate at which the update of the coefficients is done during learning and the kind of sequence of task that we want to address will jointly determine the goodness of our method.

**Algorithmic description of the method** Now we will try to give an algorithmic description of how the method can be used in learning and in inference. We will focus on a supervised learning sequence of tasks defined by a sequence of training sets $D_1, D_2, \ldots, D_T$ where $D_j = \{(x_k^j, y_k^j) : k = 1, \ldots, M_j\}$ with $x_k^j \in \mathbb{R}^d$ and $y_k^j \in \mathbb{R}^m$, $M_j$ is the dimension of the set and $m \geq 1$ is the number of classes at task $j$. We further assume that in each task we train the network using a gradient-based method for a number $E_j$ of epochs. Finally, we assume to have an *update policy* $\pi_j^\alpha$ for $j = 1, \ldots, T$ and $\alpha = 1, \ldots, E_j$ that is 1 if we want to update the approximation of the weights and 0 otherwise. Then the training process is done as described in Algorithm 1. The procedure returns for each weight $i = 1, \ldots, p$ a set of coefficients $\tilde{C}_i^S$ that express the final approximation of each weight trajectory, where $S$ is the number of Euler steps done overall.

Once the learning is completed, the output of $f$ on a new example $x$ is computed as follows:

1. we determine the task ID $j$ to which $x$ belongs and we compute the time $\bar{t}$ corresponding to the last update of $C$ within the task $j$;

2. we compute the corresponding approximated weights using equation 2, that is $w_i^*(\bar{t}) := \sum_{n=0}^{N-1} (\tilde{C}_i^S)_n v_n^{\tau S}(\bar{t})$, where $v_n^{\tau S}$ are the Legendre polynomial defined on the temporal interval $[0, \tau S]$.[3]

3. we compute the output as $f(x, w^*(\bar{t}))$

Which basically means that we use the approximation found on the whole temporal horizon of the training, which is stored in $C_i^S$ with $i = 1, \ldots, p$, to retrieve the correct weights found at the end of the corresponding task in order to make an inference on a new sample $x$.

---

[2]This can be understood since we know that $\sup_{k>0} |w_i^{k+1} - w_i^k|/\tau$, is an approximation of the Lipshitz constant of $\omega_i$ (if $\omega_i$ is $C^1$ with bounded derivative), which enter exponentially in the error bound of the Euler method (see Burden et al. (2015) Theorem 5.9 page 271).

[3]Notice that the number of steps $S$ and the time $\bar{t}$ both depends on the update policy $\pi$. For instance if the policy is $\pi_j^\alpha \equiv 1$ for all $\alpha$ and $j$ that means that the updates are done at the end of any epochs, then $S = \sum_{j=1}^T E_j$ which is the total number of epochs, but in general can be strictly less than that.

## 4 EXPERIMENTS

As we discussed in Section 3, the proposed method arises from the idea that, throughout the continual learning across many different tasks, the parameters of a NN model form a trajectory whose behaviour can be captured by an online approximation method. We remind that since the method relies on an approximation of a differential equation, we expect it would be more effective in scenarios in which the underlying trajectories of the weights have slower variations; we expect this to happen in continual learning problems in which one task is obtained from another with a *continuous shift* of data distribution. On the other hand, we still want to assess the performance of the method on more standard continual learning settings. For these reasons, we devised a set of experiments covering both cases. We will use **Incrementally Permuted MNIST**, **Incrementally Permuted Fashion MNIST** and **Incrementally Permuted Cifar10** as examples of dataset with gradual data shift, alongside with the more standard and challenging class incremental settings of **Split Cifar100** and **Split CUB200**.

In all the experiment, we trained the model by minimizing a Cross Entropy loss function within each task and using an Adam optimizer with initial learning rate of $10^{-3}$. We measured the performances of all the experiments in terms of average accuracy. Formally, let us define $a_{ij}$ to be the accuracy evaluated on the test set of task $j$ with a model that has been trained from task 1 up to task $i$ (here we are using the same notations introduced in Mai et al. (2022)). The *average accuracy* up to task $i$ $\overline{a}_i$ is then defined as $\overline{a}_i = \sum_{j=1}^{i} a_{ij}/i$, $i = 1, \ldots, T$, and we will refer to $\overline{a}_T$ as the final average accuracy. In every setting we compare the results of our method OWA with the natural baseline, which is the model trained sequentially on each task without any continual learning strategy that we will refer to as *vanilla* and with the model trained using a replay buffer strategy which we will refer to as *replay*. For a fair comparison, we always employ a buffer size that is comparable to the memory budget employed in the OWA. More in details, we employ a buffer of $n$ samples, where the size is computed as $n = (N \cdot p)/d$ where we remind that $p$ the size of a given model and with $d$ the number of features of each sample, and assuming that each weight of the model and each feature are represented using the same precision (i.e., 32 bits). The $n$ samples are drawn randomly and uniformly from each of the $T$ training distribution, i.e. with $n/T$ samples originating from each distribution. In each scenario, we tested our method under different budget that we label as *small*, *medium* and *large* and for each of these cases we compare different order of approximations with a replay method equipped with a buffer sized accordingly. All the reported numerical results have been averaged over three different runs that differs only on the initial seed with which the parameters of the network have been initialized. Additional results can be found in Appendix A.

### 4.1 INCREMENTALLY PERMUTED DATASETS

For Incrementally Permuted MNIST, Incrementally Permuted Fashion-MNIST and Incrementally Permuted Cifar10 we used OWA with the basic update policy $\pi_j^\alpha \equiv 1$ for all $\alpha = 1, \ldots, E_j$ and all $j = 1, \ldots, T$.

**Incrementally Permuted MNIST**   Is a variation of the standard Permuted MNIST continual learning benchmark ( Goodfellow et al. (2014); Kirkpatrick et al. (2017)) in which each new task is recursively generated by the previous one by permuting a fixed number of pixels. This clearly creates a shift in the distribution of the data which is softer than the one associated with the standard Permuted MNIST. To tackle this problem we used a neural network with a single hidden layer composed of 100 neurons and sigmoidal activation function. In all our experiments on this dataset we permuted 100 pixels at a time. See Appendix A for some visualizations of the incremental perturbations in the MNIST and FMNIST Incrementally Permuted datasets. We run the experiments on the datasets for $T = 100$ tasks and in each task we perform a batched optimization with a batch size of 128 for 10 epochs. We run experiments with order of approximation $N \in \{2, 10, 20\}$. The results on the final average accuracy are reported in Table 4.1 while the behaviour of the average accuracy as a function of the training steps is depicted in Figure 4.1 alongside with the baseline and the curve relative to the replay methods. As we can see from Table 4.1 the final average accuracy is always significantly higher than the baseline (+35–60%) but also of the model trained with the replay strategy (+25–34%). With $N = 10$ the accuracy is comparable with the one obtained only on the test set of the last task and when $N = 20$ it is even higher.

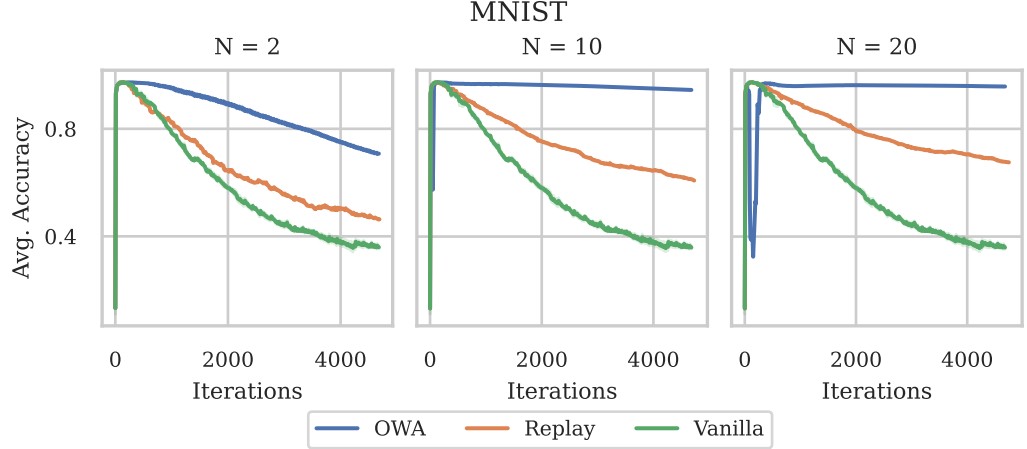

Figure 2: Incrementally Permuted MNIST: a comparison of the average accuracy of OWA against replay and a vanilla strategy for order approximation $N = 2, 10, 20$.

| dataset | vanilla | small ($N = 2$) | | medium ($N = 10$) | | large ($N = 20$) | | Single Task Accuracy |
|---|---|---|---|---|---|---|---|---|
| | | replay | ours | replay | ours | replay | ours | |
| MN | $0.36 \pm 0.01$ | $0.46 \pm 0.02$ | $\mathbf{0.71} \pm \mathbf{0.01}$ | $0.61 \pm 0.01$ | $\mathbf{0.95} \pm \varepsilon$ | $0.68 \pm 0.01$ | $\mathbf{0.96} \pm \varepsilon$ | $0.95 \pm \varepsilon$ |
| FMN | $0.31 \pm 0.02$ | $0.47 \pm 0.01$ | $\mathbf{0.61} \pm \mathbf{0.01}$ | $0.60 \pm 0.01$ | $\mathbf{0.82} \pm \varepsilon$ | $0.65 \pm \varepsilon$ | $\mathbf{0.85} \pm \varepsilon$ | $0.83 \pm \varepsilon$ |
| C10 | $0.20 \pm \varepsilon$ | $0.25 \pm 0.01$ | $\mathbf{0.40} \pm \varepsilon$ | $0.31 \pm \varepsilon$ | $\mathbf{0.49} \pm \varepsilon$ | $0.34 \pm \varepsilon$ | $\mathbf{0.50} \pm \varepsilon$ | $0.48 \pm \varepsilon$ |

Table 1: *Incrementally Permuted experiments.* Final average accuracy on the datasets Incrementally Permuted MNIST (MN), Incrementally Permuted Fashion-MNIST (FMN) and Incrementally Permuted Cifar10 (C10), reported values are in [0,1]. With $\varepsilon$ we indicate a variance smaller than $10^{-2}$. **Notice how OWA gains 20–60% points w.r.t. a vanilla strategy and 14–34% points w.r.t. replay.**

**Incrementally Permuted Fashion-MNIST**   This dataset is the analogue of the Incrementally Permuted MNIST but with the Fashion-MNIST (see Xiao et al. (2017)) dataset used as the base set and a with each new task obtained with a permutation of 100 pixels. The experimental setting is the same as for the Incrementally permuted dataset: same network and loss, $T = 100$ batch size 128, 10 epochs for each task. Again, we vary the number of term in the approximation in the set $N \in \{2, 10, 20\}$. Results are summed up again in Table 4.1. As for the MNIST dataset OWA surpasses both the baseline (+30–54%) and the replay method (+14–20%) and for $N = 20$ outperforms even the local accuracy on the last test set.

**Incrementally Permuted Cifar-10**   This dataset is the incrementally permuted version of Cifar-10 where each image is a $32 \times 32 \times 3$ image and the permutations are applied across all the image and among all the channels. Here each subsequent task has a permutation of 500 pixels with respect to the previous one. Here we are using a different model, specifically a ResNet10 architecture. We considered a sequence of $T = 100$ tasks and in each task we perform a batched optimization with a batch size of 128 for 3 epochs. In Table 4.1 we reported the results for the various order of approximations $N \in \{2, 10, 20\}$. As for the other two Incrementally Permuted datasets OWA has significantly better performances in terms of average accuracy with respect to the vanilla case (+20–28%) and also when the learning is done using replay methods (+14–18%). Again, for $N = 20$ the accuracy is on par with the local accuracy on the last test set.

### 4.2   CLASS INCREMENTAL/SPLIT DATASETS

As anticipated, the class incremental settings seems to be a challenging scenario for our method because, differently from the Incrementally Permuted one, in this scenario the domain shift between

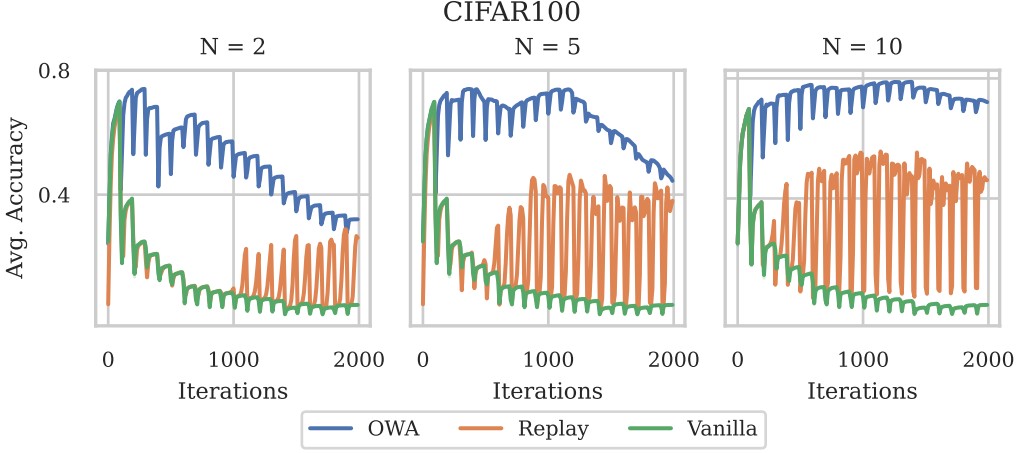

Figure 3: CIFAR100 - Split: a comparison of the average accuracy of OWA against replay and a vanilla strategy for order approximation $N = 2, 5, 10$.

different tasks is abrupt: when we require the network to learn a new set of classes, the novel training set may share very few features with the previous one and the network weights may require to change significantly. Nonetheless, since it is a commonly evaluated scenario in continual learning, we experimented on the Cifar100-Split and the CUB200-Split datasets. In both settings we employed OWA with a less frequent update policy $\pi_j^\alpha$ than previously. In particular, we heuristically found that updating 10 times per task (uniformly distributed along the epochs) resulted in an efficient update policy. In both settings we employed a Resnet18 model in transfer learning, i.e., we only trained the last fully connected layers. To do so, in Cifar100 the images have been up scaled to 224x224. The interesting results obtained in both contexts confirm the quality of the proposed approach also in this challenging scenario.

**Cifar100 - Split** The very well known Cifar100 dataset is often employed for testing Class-incremental learning settings Lopez-Paz & Ranzato (2017); Adel et al. (2019). In this work, we split the 100 classes into 20 different sets composed of 5 classes each following the semantic grouping given by the 20 super classes (Reptiles, Vehicles, Peoples, etc.). We trained the network with full batch for 100 epochs. In Figure 4.2 and in Table 4.2 we compare the performance of a model when equipped with OWA, with replay and with a vanilla learning strategy under three different budget settings. Due to the lower number of tasks ($T = 20$) we chose smaller approximation orders, i.e. $N \in \{2, 5, 10\}$. OWA in all scenarios result the best strategy with +21–67 % gain with respect to vanilla and with +6–26% with respect to replay.

**CUB200 - Split** Also CUB200 is a frequently employed dataset for Class-incremental learning settings Boschini et al. (2022). In this work, we split the 200 classes into 20 different sets composed of 10 classes each. We trained the network with a batch size of 128 samples for 100 epochs. As it can be notice in Table 4.2, also in this case the proposed model results to be still much better than a vanilla strategy (+27–82 %). However, when compared with a replay strategy, OWA results slightly less performing in lower and medium budget scenarios (-3–12%). In higher budget scenario, however, the proposed approach still improves significantly with respect to replay (+18%). We believe that this is due to the fact that the abrupt domain shift among different tasks, in this setting, can be well approximated only when using a higher order of approximation.

## 5 RELATED WORK

The present work brings together ideas that comes from the theory of online approximation of functions and continual learning. Because of the importance of the continuous/lifelong learning scenario in addressing a wide range of learning problems, there has been a significant surge in the publi-

| dataset | vanilla | small ($N = 2$) | | medium ($N = 5$) | | large ($N = 10$) | | Single Task Accuracy |
| | | replay | ours | replay | ours | replay | ours | |
| --- | --- | --- | --- | --- | --- | --- | --- | --- |
| C100 | $0.05_{\pm\varepsilon}$ | $0.26_{\pm 0.01}$ | $\mathbf{0.32}_{\pm\mathbf{0.01}}$ | $0.38_{\pm\varepsilon}$ | $\mathbf{0.44}_{\pm\varepsilon}$ | $0.46_{\pm\varepsilon}$ | $\mathbf{0.72}_{\pm\mathbf{0.01}}$ | $0.91_{\pm\varepsilon}$ |
| CB200 | $0.05_{\pm\varepsilon}$ | $\mathbf{0.48}_{\pm\mathbf{0.02}}$ | $0.45_{\pm 0.01}$ | $\mathbf{0.63}_{\pm\mathbf{0.02}}$ | $0.51_{\pm 0.02}$ | $0.69_{\pm 0.02}$ | $\mathbf{0.87}_{\pm\mathbf{0.01}}$ | $0.95_{\pm 0.02}$ |

Table 2: *Class incremental experiments.* Final average accuracy on the datasets Cifar100-Split (C100) and Incrementally Permuted CUB200 (CB200), reported values are in [0,1]. With $\varepsilon$ we indicate a variance smaller than $10^{-2}$. **Notice how also in this case OWA gains 21–82% w.r.t. a vanilla strategy and is on par with replay -12% – +18%.**

cation of scientific papers on continual learning in recent years. Many of the recently proposed methods (see the surveys of Parisi et al. (2019); De Lange et al. (2022); Mai et al. (2022); Lesort et al. (2020); Pfülb & Gepperth (2019)) falls in one of the following classes: (a) regularisation-based, (b) architecture-based also known as parameter-isolation methods and (c) memory-based or replay methods. (a) They add constraints (usually soft), to prevent too abrupt shift of the parameters between one task and the next with the purpose of learning new information while keeping as much previous knowledge (Kirkpatrick et al. (2017); Nguyen et al. (2018); Ahn et al. (2019); Zenke et al. (2017); Zhang et al. (2020)). Our method, OWA, while not falling in this category can be readily be applied in conjunction with most regularization-based methods. (b) In this family falls all the methods that uses different parameters for each new task, either by keeping the architecture fixed and storing copy of parameters (or subset of parameters) dedicated to each task or by allowing new branches associated to new tasks (Rusu et al. (2016); Xu & Zhu (2018); Aljundi et al. (2017); Fernando et al. (2017); Mallya & Lazebnik (2018); Serra et al. (2018)). While OWA cannot be strictly categorized within this class it shares with this approach the idea that one can retrieve a set of weights that is *specialized* for a specific task. (c)Here the main idea is to try to mitigate forgetting keeping in memory examples from previous tasks or to train a generator to supply pseudo-examples (Rebuffi et al. (2017); Rolnick et al. (2019); Isele & Cosgun (2018); Chaudhry et al. (2019); De Lange & Tuytelaars (2021); Lopez-Paz & Ranzato (2017); Aljundi et al. (2019); Robins (1995)). Usually in this setting as in (b) a budget need to be set to limit the amount of replay examples that one can keep in memory; in our method the choice order of approximation $N$ can be considered similarly a way to set a budget. Recently new ideas has been put forward within the field of online approximation of functions that has driven numerous new results within the machine learning community. The seminal work of Voelker et al. (2019) and the more recent paper Gu et al. (2020), proposed the main idea on online function approximation that we are using in the present paper and has inspired a whole new line of research on Deep State Models (Gu et al. (2021b;a); Smith et al. (2022); Gupta et al. (2022a;b); Gu et al. (2022)).

## 6 CONCLUSIONS

In this paper, we proposed an Online Weight Approximation method for tackling continual learning problems. Inspired by recent works on online approximation theory of scalar functions, our approach consists of an online approximation schema for all the trajectories of the weights of a network as they evolve across different tasks. More precisely OWA, given the task ID, can retrieve a task-specific weight representation that closely aligns with the locally learned parameters for that task. This allows to bring back the model to a state where it is capable to accurately process samples originating from the corresponding task distribution. In the experiments we validated the proposed approach on a variety of learning scenarios spanning over domain-incremental and class-incremental learning. We showed that the performance of a continually learnt model equipped with a sufficiently powerful OWA closes the gap with locally learnt models and surpass a strong existing continual learning competitor. In future work, we plan to test the feasibility of OWA in conjunction with methods predicting the task ID. Furthermore, we plan to test the proposed method to tackle task-free continual learning settings where not only the task ID is not available but also the task boundaries, since the proposed method is naturally posed in a continuous time setting.

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

# A ADDITIONAL RESULTS

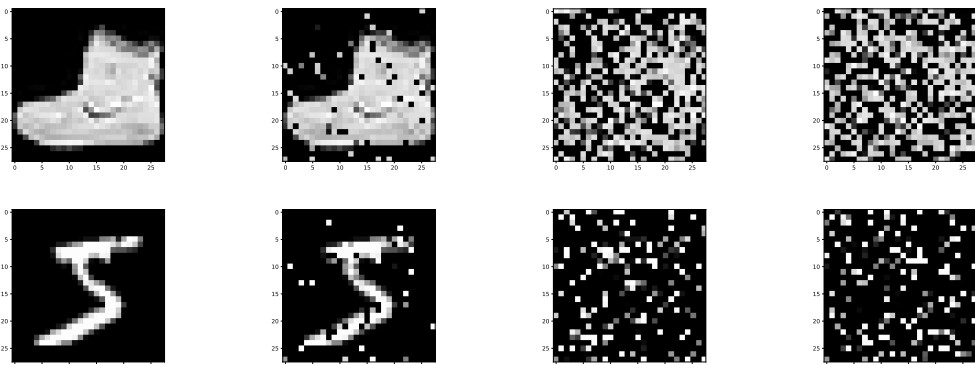

Figure 4: Same sample across different tasks $j = 1, 2, 10, 20$ in the Incremental Permuted scenarios

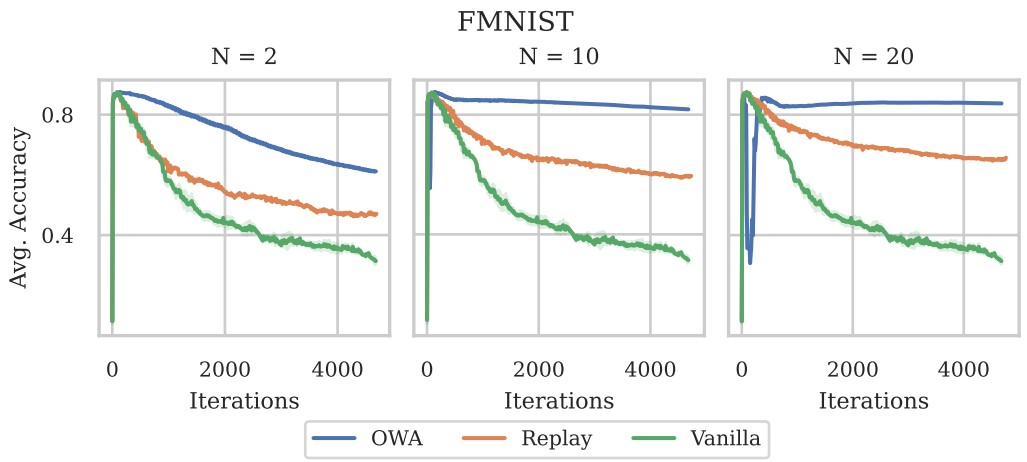

Figure 5: FMNIST: the OWAcumulative accuracy when compared against the for order approximation 2, 10, 20.

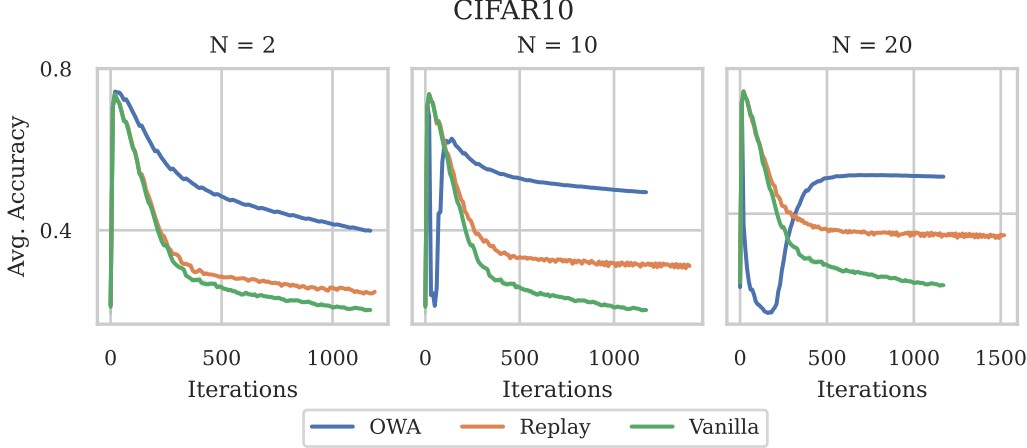

Figure 6: CIFAR10: the two baselines and then the hippo cumulative accuracy for order approximation 2, 10, 20.

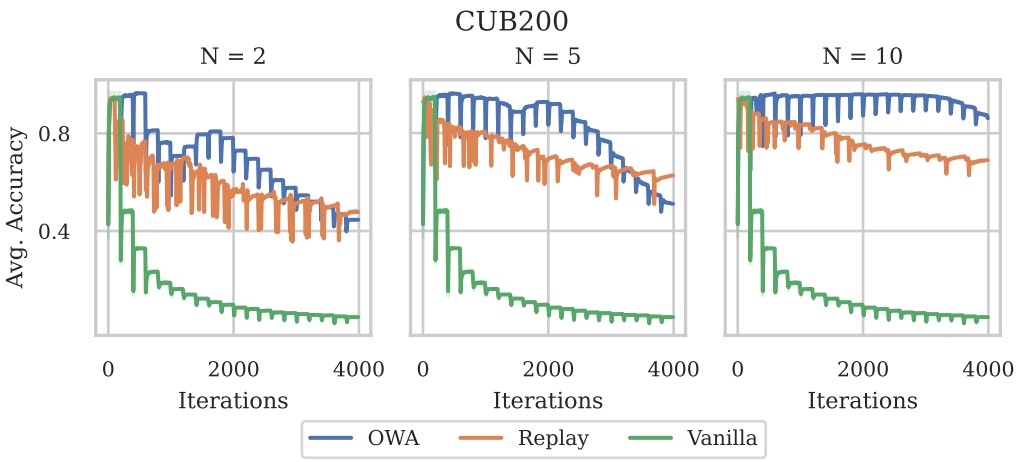

Figure 7: CUB200: the two baselines and then the hippo cumulative accuracy for order approximation 2, 10, 20.

