# OpenReview forum: "Online Weight Approximation for Continual Learning"
_ICLR.cc/2024/Conference — Submitted to ICLR 2024_

### Official Review · Reviewer_Si3p · 2023-10-19

**Soundness:** 2 fair
**Presentation:** 2 fair
**Contribution:** 2 fair
**Rating:** 3
**Confidence:** 4

**Summary:**

This paper presents Online Weight Approximation (OWA) to address the forgetting in continual learning. Specifically, OWA tries to model the dynamics of the weights across the learned task sequence to mitigate the forgetting issue. Experiments on several datasets and baselines show the effectiveness of the proposed method.

**Strengths:**

This paper proposes  Online Weight Approximation (OWA)  to mitigate forgetting in continual learning.

**Weaknesses:**

* The paper writing needs to be further improved. In the abstract section, this section should clearly state what specific problem you are trying to address. For example, the authors state that they are trying to address catastrophic forgetting. The forgetting issue is extensively studied in existing works. The abstract should state what are the limitations that exist in CL literature. Then, this paper should state how their proposed method addresses this specific problem.  Second, the motivation of introducing Online Weight Approximation (OWA) to continual learning is unclear. After reading the paper, it is unclear how OWA addresses the forgetting issue. Third, in the related work section, not only related works should be presented, but also their limitations and relation of the proposed OWA with existing methods should be clearly presented. Lastly, the method description should be clearly connected with continual learning.


* Lack of insight into how the proposed OWA mitigates forgetting. It would be better to provide more insights and illustrations of OWA and the connection with catastrophic forgetting.


* It is unclear the advantage of the proposed method compared to other related methods. In the introduction and the method sections, the paper should state the advantages of the proposed method compared to others. This would make the positions of the paper more clear.


* The compared baselines are too weak. OWA is only compared to a simple replay method. More recent memory-replay methods should be compared. For example, [1, 2] should be compared.




Reference


[1]  Dark Experience for General Continual Learning: a Strong, Simple Baseline. Neurips 2020

[2]  Gradient Projection Memory for Continual Learning.   ICLR 2021

**Questions:**

N/A

---

> ### Author Response · Authors · 2023-11-23
> **Answer to Reviewer Si3p**
>
> We thank the reviewer for their suggestion regarding the scope of the paper
> and the provided reference for further experimental comparison.
>
> Regarding the scope, we should probably specify better which particular
> target we checked in the experiment. However, we thought that our
> contribution could go beyond the tasks tested in the experimental campaign,
> since the whole continual learning community could benefit from the
> intuitions and results provided in this paper. Also, since our contribution
> do not build upon any previous method, we thought not to be fair to criticize
> the approaches of related works or highlight their limitations.
>
> The advantage of the proposed method consists in the possibility to train a
> single model on several tasks while retaining the capability to predict on
> previous tasks by means of a simple weight adaptation. The memory required
> for this adaptation is limited, being only N times the number of parameter in
> the network, where N is the number of coefficients required to approximate
> the trajectory of the weights across the task.  As we show in the
> experiments, the proposed approach is particularly efficient in scenarios
> with many tasks with respect to a simple experience replay method, which
> would require much more memory to obtain similar results.

---

### Official Review · Reviewer_nrVT · 2023-10-30

**Soundness:** 2 fair
**Presentation:** 2 fair
**Contribution:** 2 fair
**Rating:** 3
**Confidence:** 4

**Summary:**

The paper presents a method based on online weight approximation for task-incremental learning.

**Strengths:**

The proposed method is new, but I believe it is weaker than some existing methods.

**Weaknesses:**

As stated in the paper, “in this work we concentrate on scenarios where at test time the identifier of the distribution from which the sample was collected is predetermined or known in advance.” You are in effect solving the task-incremental learning (TIL) problem. Many TIL techniques can already achieve forgetting-free. Your method is also weaker than some existing methods. So, the value of your method is limited. Please check out the following,

(1)	Serra et al. Overcoming catastrophic forgetting with hard attention to the task. ICML-2018.
(2)	Wortsman et al. Supermasks in superposition. NeurIPS-2020.
(3)	Ke, et al. Achieving Forgetting Prevention and Knowledge Transfer in Continual Learning. NeurIPS-2021.
(4)	Lin et al. Trgp: Trust region gradient projection for continual learning, ICLR-2022
(5)	Lin et al. Beyond not-forgetting: Continual learning with backward knowledge transfer. NeurIP-2022.

The paper seriously lacks citations and discussion of and experimental comparisons with related literature. The systems in the above references and (6) below should be compared.

In the later part of the paper, you stated, “Throughout this paper we will assume the more definite scenario of task incremental learning.” But in the abstract and introduction, you said domain-incremental learning and class-incremental learning.

Your first set of experiments (section 4.1) should have some knowledge transfer. References (3), (4) and (5) can transfer knowledge across tasks. Your method by nature cannot perform knowledge transfer.

In your second set of experiments (section 4.2), you claimed that you are doing class-incremental learning, but since you have no task identification prediction, you are doing TIL. Your evaluation metric is average incremental accuracy, and the results are weak. Please see reference (6) below, which gives the last accuracy after all tasks are trained. The last accuracy should be much lower than average incremental accuracy. A principled way for task identification prediction is also offered in (6).

(6). Kim et al. A theoretical study on solving continual learning. NeurIPS-2022

What data is used to train eq 7? It isn’t described explicitly in the paper. My understanding is that it is just the data of each task. Then, your method is very similar to SupSup (2). The size of C^j for each task j is potentially the full network size.

Is w^i represents the set of weights for task i?

What is N is eq 3? I am unfamiliar with the order of approximation.

Why do you use ResNet10 for some datasets and ResNet18 for some other datasets?

**Questions:**

The questions are included in the weakness section.

---

> ### Author Response · Authors · 2023-11-23
> **Answer to Reviewer nrVT**
>
> We thank the reviewer for their valuable comments.
>
> However, we do not understand the comment, “Your method by nature cannot
> perform knowledge transfer”. The manner in which the coefficients
> approximating the weight trajectory among different tasks are adjusted,
> distinctly retains a memory of the previous tasks. Conversely, newer tasks
> modify the weight approximation even at the outset of the task sequence. We
> would greatly appreciate it if you could provide further clarification on
> this remark.
>
> Regarding the accuracy, what we are displaying in the plots is actually the
> accuracy on current and previous tasks after training on the current task.
> It is not the average incremental accuracy which would be, clearly, much
> higher.
>
> Regarding the data used to train Eq. (7), the update of the coefficient is
> solely based on the value of the $i$-th weight of the network at a specific
> training epoch, as outlined in Algorithm 1. The notation $w_i^k$ denotes the
> value of the $i$-th weight of the network at the $k$-th update step in our
> method. If we employ an update policy that adjusts the coefficients only once
> per task, then indeed $w_i^k$ represents the value of the $i$-th parameter of
> the network at the conclusion of the $k$-th task.
>
> The parameter $N$ represents the number of Legendre polynomials utilized in
> the approximation, as in Eq. (2). We acknowledge that our definition of this
> crucial parameter should be clearer, and we addressed it.

---

### Official Review · Reviewer_fbhj · 2023-10-31

**Soundness:** 2 fair
**Presentation:** 2 fair
**Contribution:** 2 fair
**Rating:** 3
**Confidence:** 5

**Summary:**

This paper proposes an "online weight approximation" scheme to enable continual learning in challenging domain incremental and class incremental learning setups.

**Strengths:**

1. The paper is well written, and the problem setup is mostly clear.
2. Mentioned problem setups are challenging and interesting.

**Weaknesses:**

My main concern is the scalability of the proposed method and also the limited empirical evaluations.

1. The proposed uses $O(n^{2})$ training time for each parameter of the model, which seems quite large, considering the number of parameters present in a typical deep neural network. Therefore, I believe it is necessary to compare the time required to train the proposed method with the other baselines.

2. Empirical evaluation does not include large deep neural networks. For CIFAR10 experiment, authors have used ResNet-10, however, it is not that deep. I suggest the authors to use deeper neural networks, like ResNet-34/50, Vgg-16/19, WideResNet.

3. Additionally, the datasets used for empirical evaluation is not large scale dataset. I suggest the authors to experiment with ImageNet-1k. If it is not possible due to limited resources, then try to evaluate on ImageNet-100.

4. I also think that it is not fair to reduce the number of replay samples of a rehearsal based methods such as Experience Replay, as you are model (parameters) is using very limited memory. This type of comparison is unfair to a rehearsal based methods. In that case, I would suggest the authors to not to use any memory at all and simply compare with regularization based methods such as EWC[1], MAS[2] etc.

5. Finally, this paper does not compare with recent continual learning baselines, EWC[1], MAS[2], GDumb [3], REMIND[4], DER++[5], CLS-ER[6].

[1] Kirkpatrick, James, Razvan Pascanu, Neil Rabinowitz, Joel Veness, Guillaume Desjardins, Andrei A. Rusu, Kieran Milan et al. "Overcoming catastrophic forgetting in neural networks." Proceedings of the national academy of sciences 114, no. 13 (2017): 3521-3526.

[2] Aljundi, Rahaf, Francesca Babiloni, Mohamed Elhoseiny, Marcus Rohrbach, and Tinne Tuytelaars. "Memory aware synapses: Learning what (not) to forget." In Proceedings of the European conference on computer vision (ECCV), pp. 139-154. 2018.

[3] Prabhu, Ameya, Philip HS Torr, and Puneet K. Dokania. "Gdumb: A simple approach that questions our progress in continual learning." In Computer Vision–ECCV 2020: 16th European Conference, Glasgow, UK, August 23–28, 2020, Proceedings, Part II 16, pp. 524-540. Springer International Publishing, 2020.

[4] Hayes, Tyler L., Kushal Kafle, Robik Shrestha, Manoj Acharya, and Christopher Kanan. "Remind your neural network to prevent catastrophic forgetting." In European Conference on Computer Vision, pp. 466-483. Cham: Springer International Publishing, 2020.

[5] Buzzega, Pietro, Matteo Boschini, Angelo Porrello, Davide Abati, and Simone Calderara. "Dark experience for general continual learning: a strong, simple baseline." Advances in neural information processing systems 33 (2020): 15920-15930.

[6] Arani, Elahe, Fahad Sarfraz, and Bahram Zonooz. "Learning fast, learning slow: A general continual learning method based on complementary learning system." arXiv preprint arXiv:2201.12604 (2022).

**Questions:**

Refer to the weakness section.

---

> ### Author Response · Authors · 2023-11-23
> **Answer to Reviewer fbhj**
>
> We thank the reviewer for proposing several improvements regarding the
> clarity of the paper and for providing valuable suggestions regarding the
> experimental comparisons.
>
> Our method necessitates the storage and the update of $N$ coefficients for
> each model parameter, resulting in $N$ times the number of parameters in the
> model. More precisely, $N$ represents the number of terms used in the
> expansion for parameter trajectory approximation. The update of these
> coefficients between different tasks is accomplished through linear
> computations (refer to Eq. 7) involving fixed, known matrices that are shared
> among all network parameters, making the process highly
> efficient. Experimentally, we did not observe a significant computational
> burden, and the experiments exhibited similar running times compared to the
> baselines.  In response to the reviewer's comments, incorporating a
> quantitative measure of this efficiency will undoubtedly enhance the
> article's quality.
>
> Regarding the choice on the number of replay samples, we agree that
> increasing the number of samples would have increased the performance of the
> compared replay method. However, we think that employing a number of samples
> equivalent to the memory required by our method would be the most fair
> setting.

---

### Official Review · Reviewer_AoB5 · 2023-11-01

**Soundness:** 2 fair
**Presentation:** 1 poor
**Contribution:** 2 fair
**Rating:** 3
**Confidence:** 4

**Summary:**

This paper proposes a continual learning scheme named Online Weight Approximation (OWA) motivated by the theory of online function approximation. The proposed method models the dynamics of the weights in deep neural networks and aims to alleviate catastrophic forgetting in task-incremental learning scenario. It is evaluated on multiple continual learning datasets and compared with a replay strategy, demonstrating its negligible loss of performance.

**Strengths:**

1) The proposed method focuses on the capacity problem of continual learning, which is a core and meaningful side in the long view of CL research. Their approach tries to fix the model capacity by sampling weights rather than storing it, which is creatively motivated from function approximation.
2) The method is based on a solid theory from computational mathematics. The paper also provides certain rigorous theorems for their method.

**Weaknesses:**

1. The experiment lacks diversity. The authors made the same experiment repeatedly on different datasets, which resulted in similar conclusions.
2. The datasets benchmarked by their experiment are relatively simple.
3. The experiment compared with too few and simple baselines, in which “Vanilla”is literally not a continual learning method.

**Questions:**

1.  It is  mentioned in the paper that memory budget in your method is constant. Could you explain that? Particularly, I wonder the number of Euler steps $S$  in the coefficients $\tilde{C}_i^S$ was fixed or not. If not, how does it evolve properly, without proportional increasing?
2. Which replay method did you compare with specifically? Replay methods have had a long history since the emergence of continual learning research, and their performances varied quite a lot. But the performance in your experiment is not so good, which is hard to believe as a state-of-the-art. It would be valuable to discuss how OWA compares to other methods.

---

> ### Author Response · Authors · 2023-11-23
> **Answer to Reviewer AoB5**
>
> We thank the reviewer for their specific questions, helping us to enhance the
> presentation of our work.
>
> Our method relies on the concept of approximating the trajectory of weights
> across different tasks. We achieve this by introducing a technique to update
> the coefficients of a specific polynomial expansion for each weight in the
> neural network (NN). Once we determine the order of the approximation,
> denoted as $N$ (representing the number of coefficients to retain in the
> expansion presented in Eq. 2), irrespective of the Euler's step performed,
> only $N$ values need to be stored in memory for each weight of the
> network. The elegance of the method lies in the fact that we simply
> *update* these coefficients rather than adding new ones. The update is
> performed in a manner that preserves the memory of the information observed.
>
> Regarding the replay strategy, we employed a straightforward experience
> replay method, determining the size of the buffer as outlined in the paper.

---

### Author Response · Authors · 2023-11-23

We would like to thank the reviewers for their valuable comments.

A recurring critique across all the reviews has been regarding the
appropriateness of the experimental evaluation.  This shows that we have not
been able to communicate effectively the reasons behind the design choices
made for the experimental set-up. We wish to address this issue more
generally in the present document.

As mentioned in the paper, we anticipate our method to be particularly
effective when the sequence of tasks we consider exhibits some form of
“continuity” in the evolution of the tasks, such as originating from a smooth
shift in the distribution of the data.  Long-term patient follow-up, with
regular medical examinations, is a real case for a such setting. While we
acknowledge the artificial nature of the datasets we have built to match with
this setting, we were not able to identify more complex and realistic
datasets suitable for meaningful evaluation. Indeed, it is a current focus of
our research to curate benchmarks that align with these criteria.

In addition to the incrementally permuted datasets, we presented results on
more classical datasets such as Split CIFAR or Split CUB. As discussed in the
paper, we find it interesting to observe that even in scenarios significantly
different from the setting for which the method was originally developed, our
approach demonstrates noteworthy performance.

---

### Meta-Review · Area_Chair_AjDi · 2023-12-10

**Metareview:**

While the proposed method is interesting, all reviewers found that the experimental validation was insufficient and/or not supporting the claims. The gains compared to published methods was also put into question. Overall, this work is not ready for publication. I believe the reviewers suggested avenues for improvements and I would encourage the authors to reflect on those.

**Justification For Why Not Higher Score:**

All reviewers agreed this paper should not be accepted (score of 3) due to a lack of supporting evidence.

**Justification For Why Not Lower Score:**

N/A

---

### Decision · Program_Chairs · 2024-01-16

Reject